# Comparison of chloroplast genomes and phylogenomics in the *Ficus sarmentosa* complex (Moraceae)

**Zhen Zhang**[1], **De-Shun Zhang**[1], **Lu Zou**[2], **Chi-Yuan Yao**[1]*

**1** College of Architecture and Urban Planning, Tongji University, Shanghai, China, **2** School of Life Sciences, East China Normal University, Shanghai, China

* cyyao@tongji.edu.cn

## Abstract

Due to maternal inheritance and minimal rearrangement, the chloroplast genome is an important genetic resource for evolutionary studies. However, the evolutionary dynamics and phylogenetic performance of chloroplast genomes in closely related species are poorly characterized, particularly in taxonomically complex and species-rich groups. The taxonomically unresolved *Ficus sarmentosa* species complex (Moraceae) comprises approximately 20 taxa with unclear genetic background. In this study, we explored the evolutionary dynamics, hotspot loci, and phylogenetic performance of thirteen chloroplast genomes (including eleven newly obtained and two downloaded from NCBI) representing the *F. sarmentosa* complex. Their sequence lengths, IR boundaries, repeat sequences, and codon usage were compared. Both sequence length and IR boundaries were found to be highly conserved. All four categories of long repeat sequences were found across all 13 chloroplast genomes, with palindromic and forward sequences being the most common. The number of simple sequence repeat (SSR) loci varied from 175 (*F. dinganensis* and *F. howii*) to 190 (*F. polynervis*), with the dinucleotide motif appearing the most frequently. Relative synonymous codon usage (RSCU) analysis indicated that codons ending with A/T were prior to those ending with C/T. The majority of coding sequence regions were found to have undergone negative selection with the exception of ten genes (*accD*, *clpP*, *ndhK*, *rbcL*, *rpl20*, *rpl22*, *rpl23*, *rpoC1*, *rps15*, and *rps4*) which exhibited potential positive selective signatures. Five hypervariable genic regions (*rps15*, *ycf1*, *rpoA*, *ndhF*, and *rpl22*) and five hypervariable intergenic regions (*trnH-GUG-psbA*, *rpl32-trnL-UAG*, *psbZ-trnG-GCC*, *trnK-UUU-rps*16 and *ndhF-rpl32*) were identified. Overall, phylogenomic analysis based on 123 *Ficus* chloroplast genomes showed promise for studying the evolutionary relationships in *Ficus*, despite cytonuclear discordance. Furthermore, based on the phylogenetic performance of the *F. sarmentosa* complex and *F. auriculata* complex, the chloroplast genome also exhibited a promising phylogenetic resolution in closely related species.

**Data Availability Statement:** The eleven newly obtained complete chloroplast genome sequences that support the findings in the study are deposited in the NCBI with accession numbers as followed: ON257105-ON257115. The corresponding high

throughput Illumina sequencing reads are deposited to Genome Sequence Archive (GSA) in BIG Data Center (https://bigd.big.ac.cn/gsa/) under BioProject accession: PRJCA012975.

**Funding:** This research was funded by National Natural Science Foundation of China from Dr. De-Shun Zhang (grant number 32071824). The funder De-Shun Zhang implemented the conceptualization, visualization, and draft review of the study.

**Competing interests:** The authors have declared that no competing interests exist.

## Introduction

The genus *Ficus* L. (Moraceae) is a species-rich taxon which contains at least 800 species and is widely distributed across tropical and subtropical regions [1–3]. Due to insufficient genetic differentiation, the genus *Ficus* is taxonomically complex and contains many sympatric species, including the *F. pedunculosa* group, *F. punctata* group, *F. chartacea* group, and *F. subulata* group, among others [1, 3, 4]. Meanwhile, widely not-strict one-to-one obligate mutualism between fig trees and fig wasps has resulted in frequent hybridization and introgression among *Ficus* species [5–8], which has so far hindered research on the taxonomy and evolutionary history [9]. Our current understanding of the *Ficus* phylogenetic framework is the result of research on a few nuclear loci, such as ITS, ETS, *G3pdh*, GBSSI, and *waxy* [3, 4, 10–14]. Although nuclear genome data have been used to reconstruct the *Ficus* phylogeny [15–17], these studies represent less than ten percent of *Ficus* species, which is unlikely to accurately represent the evolutionary relationship in the genus *Ficus*.

While the use of nuclear genome data has advantages for the detection of hybridization and introgression, organellar genomic resources are also of great importance to evolutionary research due to maternal inheritance as a single unit [18, 19]. In the last decade, due to the rapidly decreasing cost of whole-genome sequencing (WGS) and the development of chloroplast genomic assembling pipelines, such as GerOrganelle [20], Fast-Plast (https://github.com/mrmckain/Fast-Plast), NOVOPlasty [21], and ORG.asm (https://git.metabarcoding.org/org-asm/org-asm), plastome-based evolutionary research has become easier and more cost-effective [22–24]. However, chloroplast genomes are publicly available for less than five percent of species in *Ficus* [25]. Bruun-Lund et al. [26] published a novel and innovative *Ficus* phylogenetic framework based on 59 newly obtained chloroplast genomes, and the results of which were obviously inconsistent with phylogenies based on the nuclear genome. Unfortunately, many of the genome sequences used by Bruun-Lund et al. contained gaps (an average of 7% missing data with a maximum of 65%), leading to difficulty in comparative chloroplast genomics and possible phylogenetic artifacts [27]. Even so, the chloroplast framework of Brunn-Lund et al. has been replicated by subsequent research generally with an extended dataset [17]. Overall, according to studies in both *Ficus* and the other taxa [28–30], it has been verified to be effective for chloroplast genomes to reconstruct the phylogenetic reference at infra-, intergeneric, and higher ranks. However, little is known about the phylogenetic performance of complete chloroplast genomes in studies of closely related species or even infra-species, particularly in the taxonomically complex genera such as *Ficus*.

The *Ficus* subg. *Synoecia* sect. *Rhizocladus* subsect. *Plagiostigma* comprises approximately 10 species and 11 varieties which are widely distributed across east Asia, parapatric to the distribution center of *Ficus* southeast Asia [1, 3, 13, 16, 31]. Subsect. *Plagiostigma* is genetically unclear and taxonomically unresolved, forming the *Ficus sarmentosa* species complex [4]. After the first systematic treatment of the *F. sarmentosa* complex by Corner in 1960 and 1965 [32, 33], few studies have attempted to unravel the taxonomic complexity of this group with the exception of descriptions of several controversial taxa in the 1980s [1, 34–37] (such as *F. dinganensis*, *F. guizhouensis*, and *F. polynervis*) and the rank elevation of some varieties (such as *F. pubigera* var. *anserina* and *F. sarmentosa* var. *thunbergii*) [38–40]. More recently, some of this phylogenetic ambiguity was resolved through the molecular work of Zhang et al. [4]. Zhang's work resulted in the rank elevation of *F. pubigera* var. *anserina* and the discovery that *F. sarmentosa* is not monophyletic, complicating the relationship between the complex and these previously described species in the 1980s. However, because only three loci (ITS, ETS, and *G3pdh*) were used to resolve the genetic background of the complex, the results were neither stable nor highly resolved [4]. The inclusion of more variable genetic loci or genome data

should be helpful to resolve the taxonomic uncertainty of the *F. sarmentosa* complex. In this study, we utilized both comparative chloroplast genomics and phylogenomics to characterize 1) the diversity of hotspot loci, 2) the variation among chloroplast genomes, and 3) the potential of chloroplast genomes to resolve the evolutionary relationships between closely related species of the F. sarmentosa species complex.

## Materials and methods

### Sample collection, DNA extraction and resequencing, and genome assembly and annotation

Healthy, young leaves were collected from the field in 2015–2021, and each was sealed in silica gel. Based on the phylogenetic relationships outlined in Zhang et al. [4], we sampled eleven taxa within the *F. sarmentosa* species complex in order to maximize genetic coverage. Detailed sample information is shown in S1 Table. All voucher specimens were stored at the herbarium of East China Normal University (HSNU).

Total DNA was extracted from 100 mg of dry leaf tissue using the CTAB method [41]. After quality detection with NanoDrop and Qubit 2.0, purified DNA samples were randomly ultrasonicated into ~350 bp segments, which were subsequently used to construct paired-end libraries. Whole-genome resequencing (WGS) was carried out using the Illumina NovaSeq 6000 platform, according to the PE150 sequencing strategy. Raw reads were filtered and cleaned according to the following criteria: reads containing > 10% unidentified nucleotides, > 50% low-quality bases (Q≤5), or adapter sequences were omitted for further analyses. Finally, the chloroplast genome sequences of two more taxa in the *F. sarmentosa* complex, *F. sarmentosa* var. *henryi* (GenBank accession no. OL415083) and *F. dinganensis* (GenBank accession no. OK375500), were included for further comparative analyses.

Clean data were used to assemble the complete circular chloroplast genome with GetOrganelle v1.6.4 [20], utilizing the "embplant_pt" model. In the case that the output sequence was not circular, the R (number of runs) and w (word size) parameters were adjusted until circularity was achieved. Chloroplast genome annotation was carried out using PGA [42], with the default parameters. The chloroplast genome was visualized using the online OGDRAW tool (https://chlorobox.mpimp-golm.mpg.de/OGDraw.html) [43]. For consistency, the two supplementary plastome sequences (OL415083 and OK375500) were re-annotated according to the same routine.

### Analysis of chloroplast genome structure

All 13 *F. sarmentosa* complex genomes were analyzed to determine the lengths and GC contents of the whole genomes, four quadripartite regions (large single copy (LSC), small single copy (SSC), and two inverted repeats (IRs)), and coding sequence (CDS) regions.

### Analysis of IR contraction and expansion

The online R Shiny application IRscope (https://irscope.shinyapps.io/irapp/) [44] was used to examine and visualize the boundary variation of LSC/IR/SSC of all 13 *F. sarmentosa* complex genomes.

### Analysis of long repeat sequences and SSRs

The four categories of long repeat sequences, forward (F), reverse (R), complement (C), and palindromic (P), were analyzed using the REPuter online tool (https://bibiserv.cebitec.uni-bielefeld.de/reputer) [45], with 50 maximum computed repeats and a minimal repeat size of 8.

One of the inverted repeat regions (IRb) was removed in the REPuter analysis to avoid repeatable results. Simple sequence repeats (SSRs) were detected using MISA (https://webblast.ipk-gatersleben.de/misa/) [46], with the following parameters: 8 repeat units for mononucleotide SSRs, 5 repeat units for dinucleotide SSRs, 4 repeat units for trinucleotide SSRs, and 3 repeat units for tetra-, penta-, and hexanucleotide SSRs. The maximal sequence length between two SSRs was set to 100 bp.

## Comparison of complete chloroplast genomes and diversity hotspot analysis

All 13 *F. sarmentosa* complex genomes were compared using the mVISTA online tool (https://genome.lbl.gov/vista/mvista/submit.shtml) [47], with the global multiple alignment model (LAGAN). The *F. anserine* chloroplast genome was used as the reference and the RankVISTA probability threshold was set to 0.5. All genes and intergenic regions were extracted from the Genbank annotation files in batches using Perl scripts created by Xiao-Jian Qu (https://github.com/quxiaojian/Bioinformatic_Scripts). Alignments of the genic and intergenic loci were carried out using MUSCLE v5.1 [48], with default parameters. After alignment, the nucleotide diversity (π) was calculated for all genic and intergenic loci.

## Codon usage analysis

To compare codon usage patterns of all the CDS sequences across 13 *F. sarmentosa* complex genomes, the relative synonymous codon usage (RSCU) was calculated using DAMBE v7.3.11 [49].

## Selective pressure analysis

The ratio of nonsynonymous (Ka) to synonymous (Ks) substitution can be used to quantify evolutionary selective pressure. The Ka/Ks ratio was calculated for all 79 unique CDSs using TBtools v1.09876 [50]. Positive selection was indicated by Ka/Ks > 1, negative (purifying) selection was indicated by Ka/Ks < 1, and neutral selection was indicated by Ka/Ks = 1. Based on the phylogenetic framework outlined in Zhang et al. [4], *F. simplicissima* was chosen as the reference to calculate Ka and Ks between *F. simplicissima* and our sampled representatives of the *F. sarmentosa* complex. For visualization purposes, the NaN value (i.e., both Ks and Ka = 0) was manually set as "1" to denote neutral selection. Finally, the infinity value (Ks > 0 and Ka = 0) was counted alone.

## Phylogenetic analysis

The 13 *F. sarmentosa* complex chloroplast genomes were combined with 36 *Ficus* chloroplast genomes from GenBank and 59 *Ficus* genomes published by Bruun-Lund et al. [26]. Additionally, 18 genomes from the China National GeneBank (accession number: CNP0001337) and Genome Sequence Archive (accession number: PRJCA002187) [15, 51], including 8 samples belonging to the *F. sarmentosa* complex, were assembled to further explore the *Ficus* phylogenomics and the potential of chloroplast genomes to resolve the evolutionary relationships among the closely related species. Seven samples from the Olmedieae tribe were chosen as the outgroup, according to previous studies [26, 52]. After discarding four samples found to be more than half (> 50%) missing data (three in the genus *Ficus* and one in the outgroup), a total of 129 chloroplast genomes were used to construct the phylogenetic tree (detailed samples information shown in S1 Table). Additionally, one of the two IRs was removed. The 129 genomes were aligned using MAFFT v7.490 [53], with the "auto" model. The aligned

sequences were trimmed with trimAl and sites with $> 10\%$ gaps were removed [54]. The maximum likelihood (ML) tree was constructed using IQ-TREE 2 with ultrafast bootstrap (-bb) and aLRT test (-alrt) numbers set at 10000. The optimal nucleotide substitution model was chosen with ModelFinder [55].

## Results

### Summary of all 13 complete chloroplast genomes in the *F. sarmentosa* complex

For the eleven newly resequenced samples, a total of 79G bases were obtained from 19,379,830 (*F. sarmentosa* var. *sarmentosa*) to 23,945,593 (*F. sarmentosa* var. *henryi*) clean paired-end reads. The ratios of chloroplast paired reads to whole reads ranged from 0.87% (*F. sarmentosa* var. *henryi*) to 4.05% (*F. guizhouensis*). The average kmer-coverage values ranged from 41.5 (*F. sarmentosa* var. *henryi*) to 128.2 (*F. sarmentosa* var. *thunbergii* and *F. pubigera*). Detailed information on high-throughput sequencing data can be found in S2 Table.

All *F. sarmentosa* complex genomes, including 11 newly obtained genomes and two genomes downloaded from GenBank, exhibited a typical quadripartite structure (Fig 1), containing one LSC, one SSC, and two IRs. The lengths of the complete chloroplast genomes ranged from 160,018 (*F. dinganensis*) to 160,385 bp (*F. sarmentosa* var. *lacrymans*) (Table 1). The LSCs accounted for 55.12–55.20% of the total genome size and ranged from 88,200 (*F. dinganensis*) to 88,535 bp (*F. sarmentosa* var. *lacrymans*) in length. The SSCs accounted for 12.51–12.55% of the total genome size and ranged from 20,064 (*F. sarmentosa* var. *lacrymans*) to 20,108 bp (*F. pubigera*) in length. The IRs accounted for 16.14–16.17% of the total genome size and varied from 25,866 (*F. dinganensis*) to 25,898 bp (*F. sarmentosa* var. *nipponica* and *F. sarmentosa* var. *thunbergii*). The CDS regions ranged from 79,149 (*F. dinganensis*) to 79,308 bp (*F. sarmentosa* var. *impressa*) in accumulated lengths. The GC content of the whole genome ranged from 35.94 to 35.99%, with *F. sarmentosa* var. *henryi* having the highest GC content, and *F. anserine* and *F. pubigera* having the lowest. The GC contents of LSCs, SSCs, and IRs were also calculated. Overall, the IRs had the highest GC content, which ranged from 42.63% (*F. anserina*, *F. howii*, and *F. pubigera*) to 42.65% (*F. pumila*, *F. sarmentosa* var. *lacrymans*, and *F. sarmentosa* var. *sarmentosa*). SSCs had the lowest GC content, which ranged from 28.94% (*F. sarmentosa* var. *lacrymans* and *F. sarmentosa* var. *sarmentosa*) to 29.09% (*F. sarmentosa* var. *henryi*, *F. sarmentosa* var. *impressa*, and *F. sarmentosa* var. *nipponica*). The same analyses applied to all 13 genomes, 131 genes were annotated, including 86 coding genes, 37 transfer RNA (tRNA) genes, and 8 ribosomal RNA (rRNA) genes. Finally, it was discovered that the *infA* gene was intensively pseudogenized [17].

### IR contraction and expansion

Across all 13 *F. sarmentosa* complex genomes, four junctions among IR, LSC, and SSC regions were compared (Fig 2). Invariably and consistently across all 13 genomes, the boundary between the LSC and IRb regions (JLB) was located within the *rps19* gene, 171 bp away from its starting base and 108 bp away from its ending base. The boundary between the IRb and SSC regions (JSB) was located within the *ndhF* gene, either 17 or 25 bp (*F. pumila*) away from its starting base and 2,236 (*F. anserina*, *F. pubigera*, *F. sarmentosa* var. *henryi*, and *F. sarmentosa* var. *sarmentosa*), 2,237 (*F. pumila*), or 2,245 bp (all other taxa) away from its ending base. The boundary between the SSC and IRa regions (JSA) was located within the *ycf1* gene, either 4,713, 4,722, or 4,724 bp away from its starting base and 1,024 (*F. pumila*) or 1,026 bp away from its ending base. For the IRa and LSC regions, the junction between them was located

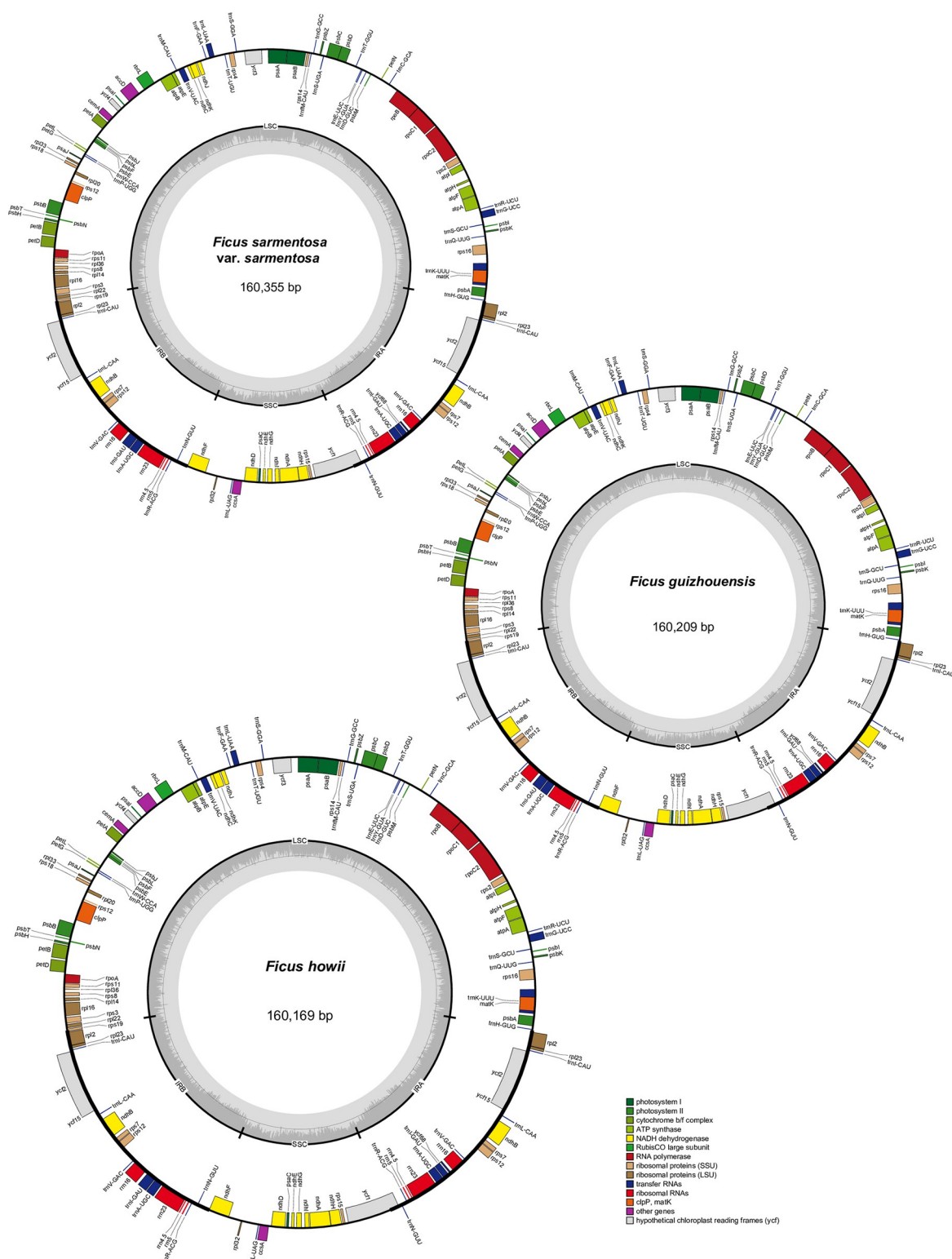

**Fig 1. Chloroplast gene maps of *Ficus sarmentosa* var. *sarmentosa*, *Ficus guizhouensis*, and *Ficus howii*.** Genes drawn inside are transcribed clockwise and genes drawn outside are counterclockwise. Genes belonging to different functional groups are color coded. In the inner circle, dark gray and light gray indicate the GC content and AT content, respectively. The boundaries of the large single copy (LSC), small single copy (SSC), and two inverted regions (IRa, IRb) are also shown in the inner circle.

**Table 1. Summary of complete chloroplast genomes of all thirteen taxa in the *F. sarmentosa* species complex.**

| Taxon | Total Length (bp) | LSC Length (bp) | SLC Length (bp) | IR Length (bp) | CDS Length (bp) | Total GC (%) | LSC GC (%) | SSC GC (%) | IR GC (%) |
|---|---|---|---|---|---|---|---|---|---|
| *F. anserina* | 160267 | 88395 | 20106 | 25883 | 79173 | 35.94 | 33.61 | 28.97 | 42.63 |
| *F. dinganensis* | 160018 | 88200 | 20086 | 25866 | 79149 | 35.98 | 33.67 | 29.00 | 42.64 |
| *F. guizhouensis* | 160209 | 88330 | 20103 | 25888 | 79272 | 35.98 | 33.66 | 29.02 | 42.64 |
| *F. howii* | 160169 | 88328 | 20076 | 25882 | 79167 | 35.98 | 33.65 | 29.03 | 42.63 |
| *F. polynervis* | 160356 | 88507 | 20063 | 25893 | 79155 | 35.95 | 33.62 | 28.95 | 42.64 |
| *F. pubigera* | 160299 | 88425 | 20108 | 25883 | 79167 | 35.94 | 33.61 | 28.98 | 42.63 |
| *F. pumila* | 160229 | 88369 | 20100 | 25880 | 79152 | 35.98 | 33.64 | 29.08 | 42.65 |
| *F. sarmentosa* var. *henryi* | 160183 | 88307 | 20080 | 25898 | 79170 | 35.99 | 33.66 | 29.09 | 42.64 |
| *F. sarmentosa* var. *impressa* | 160293 | 88417 | 20080 | 25898 | 79308 | 35.97 | 33.63 | 29.09 | 42.64 |
| *F. sarmentosa* var. *lacrymans* | 160385 | 88535 | 20064 | 25893 | 79155 | 35.95 | 33.63 | 28.94 | 42.65 |
| *F. sarmentosa* var. *nipponica* | 160314 | 88436 | 20082 | 25898 | 79170 | 35.97 | 33.63 | 29.09 | 42.64 |
| *F. sarmentosa* var. *sarmentosa* | 160355 | 88501 | 20068 | 25893 | 79155 | 35.95 | 33.62 | 28.94 | 42.65 |
| *F. sarmentosa* var. *thunbergii* | 160282 | 88397 | 20089 | 25898 | 79179 | 35.98 | 33.65 | 29.06 | 42.64 |

between the *rpl2* and *trnH* genes, either 62 or 63 bp away from the starting base of the *trnH* gene.

## Long repeat sequences and simple sequence repeats (SSRs)

Across all 13 *F. sarmentosa* complex genomes, a total of 373 long repeat sequences were identified, representing all four repeat categories: forward (F), reverse (R), complement (C), and palindromic (P) (Fig 3A). All four categories of repeats were detected within all 13 chloroplast genomes. Four taxa, *F. anserina*, *F. dinganensis*, *F. polynervis*, and *F. sarmentosa* var. *sarmentosa*, contained the greatest number of long repeat sequences (31), while *F. sarmentosa* var. *thunbergii* contained the least (26). Among all four repeats, P repeats were the most common across all 13 genomes, ranging from 14 (*F. guizhouensis*) to 16. There were relatively fewer R and C repeats, with *F. polynervis*, *F. sarmentosa* var. *lacrymans*, and *F. sarmentosa* var. *sarmentosa* containing the most (4). For the lengths of long repeat sequences, 30–39 bp is the most common ranging from 21 (*F. sarmentosa* var. *thunbergii*) to 26 (*F. anserina* and *F. dinganensis*) times (Fig 3B). The long repeat sequences over 60 bp in length were the least, appearing only once. The maximum length among all the long repeat sequences was 64 bp.

Simple sequence repeats (SSRs) consisting of the 1- to 6- nucleotide motifs were surveyed across the 13 *F. sarmentosa* complex genomes (Fig 4A). Mononucleotide SSRs were the most abundant (78.71%) across all 13 genomes. Dinucleotide SSRs, the most commonly used motif in population genetics and phylogenetics, were also relatively abundant, appearing from 19 (*F. howii*, *F. polynervis*, *F. sarmentosa* var. *lacrymans*, and *F. sarmentosa* var. *sarmentosa*) to 22 (*F. pumila* and *F. sarmentosa* var. *thunbergii*) times. It is noteworthy that tetranucleotide SSRs were more common than trinucleotide SSRs (Fig 4A), considering the former appeared from 9 (*F. guizhouensis*, *F. polynervis*, *F. sarmentosa* var. *lacrymans*, and *F. sarmentosa* var. *sarmentosa*) to 11 (*F. pumila*, *F. sarmentosa* var. *henryi*, *F. sarmentosa* var. *impressa*, and *F. sarmentosa* var. *nipponica*) times. Hexanucleotide SSRs were absent from all genomes. The total number of SSRs varied from 175 (*F. dinganensis* and *F. howii*) to 190 (*F. polynervis*). Of the repeat

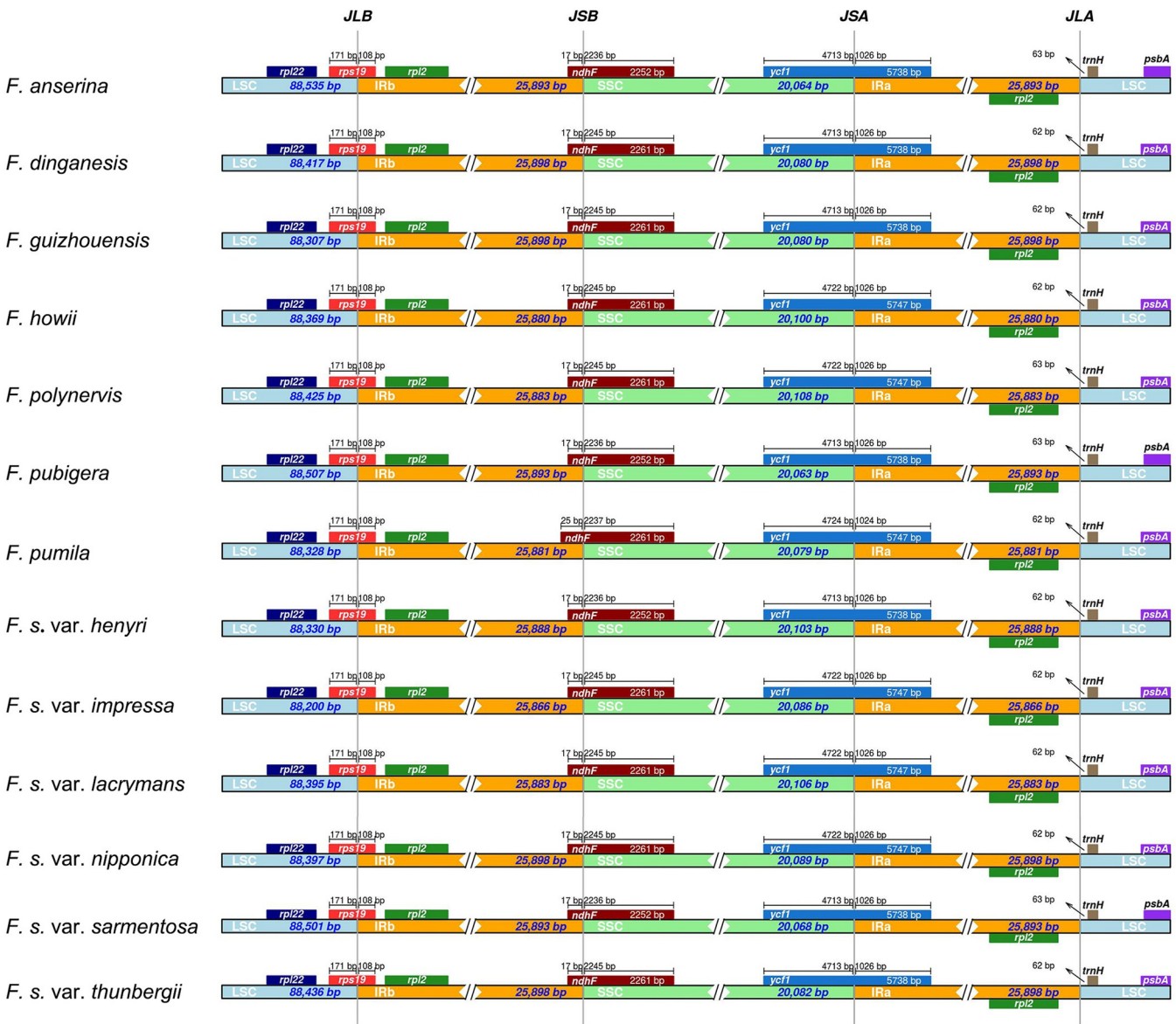

**Fig 2. Comparison of the boundaries between the large single copy (LSC), small single copy (SSC), and two inverted repeat regions (IRs) among 13 chloroplast genomes in the *F. sarmentosa* complex.** The numbers around the vertical lines indicate the distances between the boundaries and the starting or ending bases of their nearest genes.

motifs, A/T was the most common mononucleotide motif, appearing from 132 (*F. guizhouensis*) to 149 (*F. polynervis* and *F. sarmentosa* var. *sarmentosa*) times (Fig 4B). AT/AT was the second most common motif, appearing between 18 (*F. howii*, *F. polynervis*, *F. sarmentosa* var. *lacrymans*, and *F. sarmentosa* var. *sarmentosa*) and 21 (*F. pumila* and *F. sarmentosa* var. *thunbergii*) times. The majority of the remaining motifs appeared only once to four times, with the exception of AATT/AATT appearing up to seven times (Fig 4B).

## Codon usage

Across all 13 *F. sarmentosa* complex genomes, the 79 unique protein-coding CDS regions were encoded by between 22,867 (*F. polynervis* and *F. sarmentosa* var. *lacrymans*) and 22,918 (*F.*

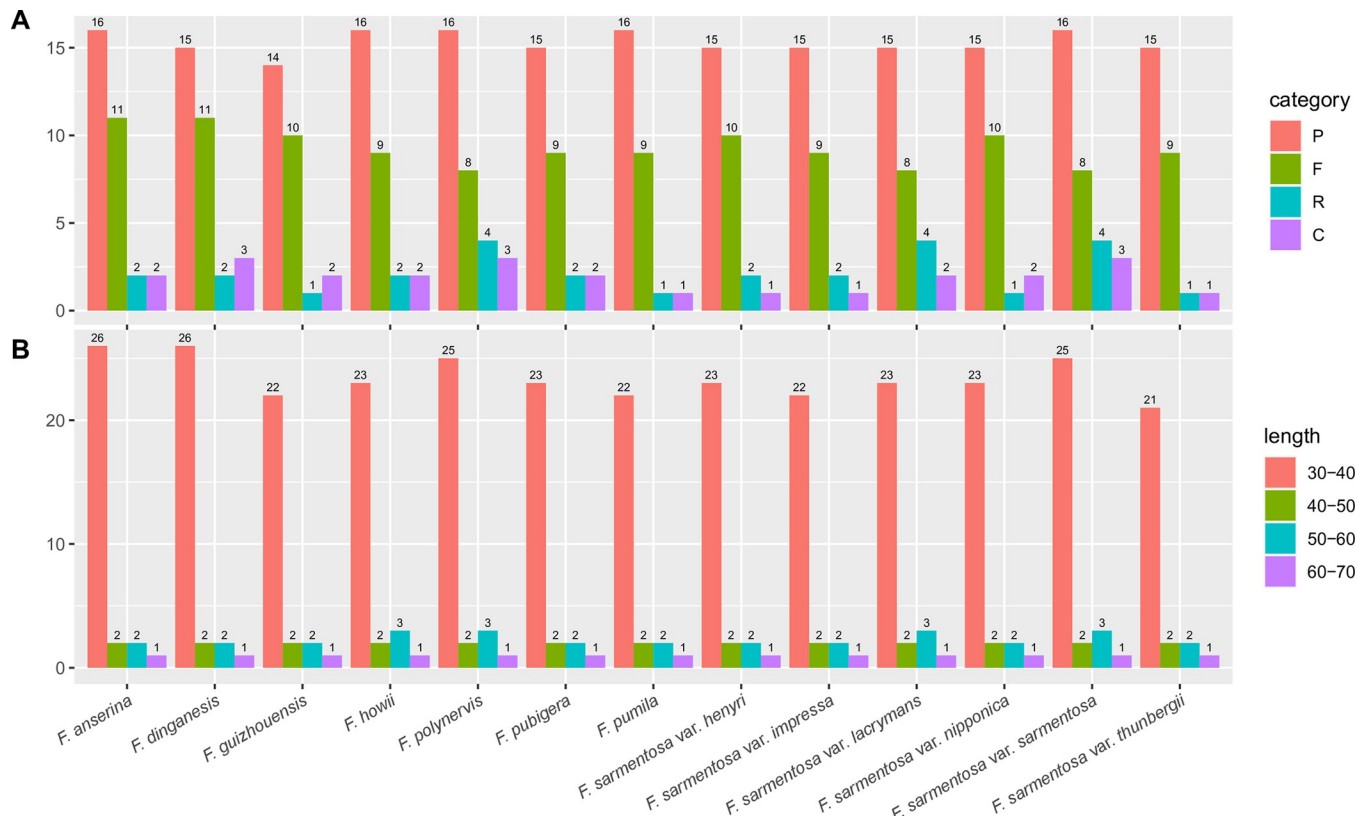

**Fig 3. Comparison of long repeat sequences among 13 *F. sarmentosa* complex genomes.** A, The number of each of four long repeat types (P, palindromic; F, forward; R; reverse; C complement); B, The number of long repeat sequences of different lengths.

*sarmentosa* var. *impressa*) codons. The codon usage among all 81 protein-coding genes is summarized in Table 2. Among these codons, CAU encoding histidine (H) was the most frequent, appearing 12,800 times across all 13 taxa. Except the stop codon, UGU encoding L-Cysteine (C) was the next rarest, appearing 718 times only. According to the RSCU analysis, GCU and CUU had the highest average values of 1.858 and 1.818, respectively (Table 2), whereas UAC and CGC had the lowest average values of 0.364 and 0.400, respectively. Among all three stop codons, UAA was the most common (53.64%). Thirty out of the 64 codons with RSCU > 1 ended with either A or U, while 32 out of 64 codons with RSCU < 1 ended with either G or C, with the exception of the AUA codon.

## Genomic divergence and hotspot regions

The divergence of whole sequence among the 13 *F. sarmentosa* complex genomes was analyzed using the mVISTA online platform with *F. anserina* as a reference. The results showed that the full-length chloroplast genomes were largely conserved across all 13 taxa. The majority of variable sites were located in intergenic spacer regions (marked red in Fig 5). Interestingly, IRs were found to be more conserved than either LSCs or SSCs.

Nucleotide diversity (π) was calculated for each gene and intergenic region to evaluate genetic differentiation and detect hyper-variable segments. Among 60 genes > 200 bp in length, *rps15*, *ycf1*, *rpoA*, *ndhF*, and *rpl22* exhibited the highest nucleotide diversity: 0.00535, 0.00462, 0.00456, 0.00427, and 0.00377, respectively (Fig 6). The alignment lengths of these

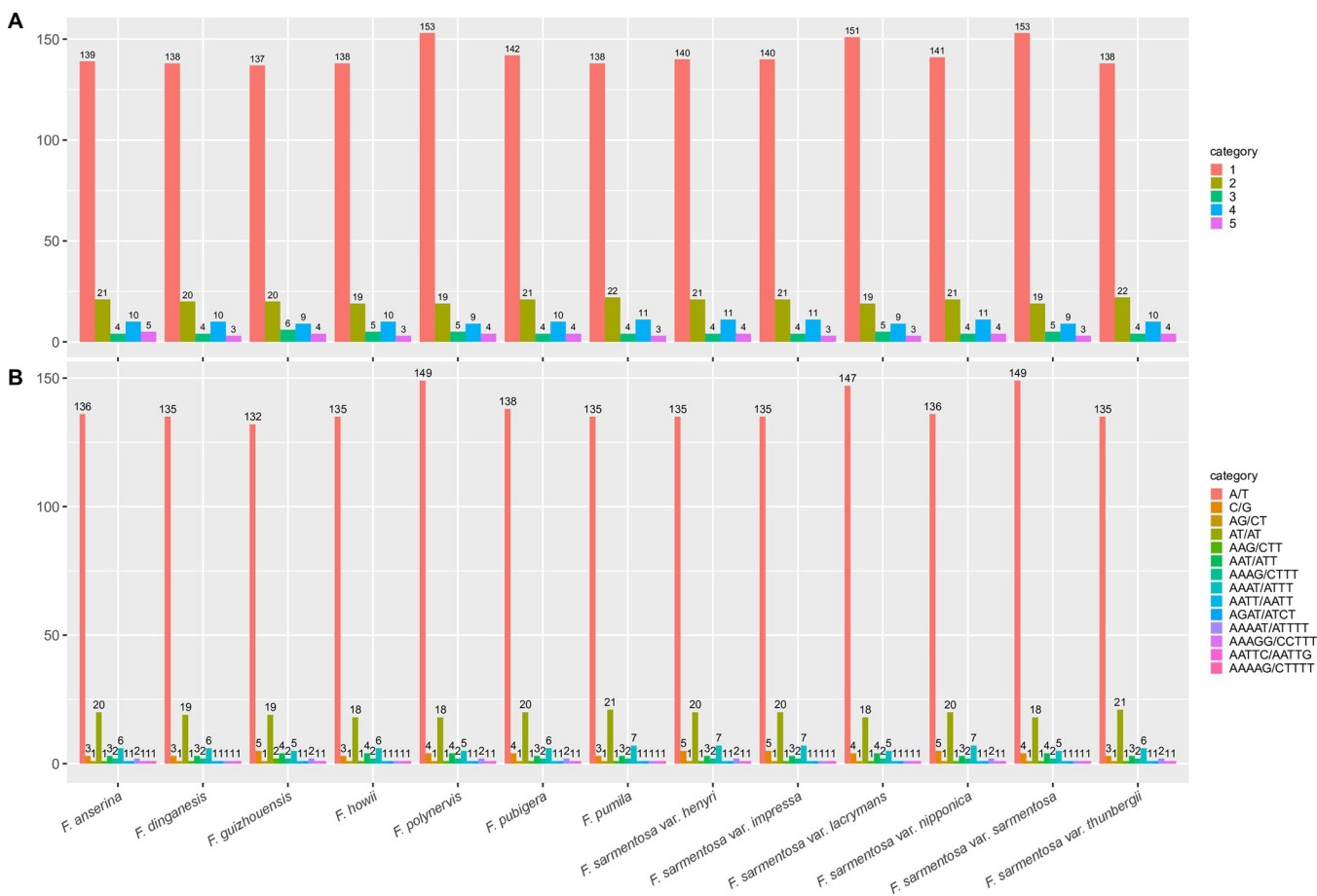

**Fig 4. Comparison of simple sequence repeats (SSRs) among 13 _F. sarmentosa_ complex genomes.** A, The number of SSRs containing one- to five-nucleotide motifs; B, The number of different SSR motifs.

five genes ranged from 282 to 5,748 bp. Seven genes (_atpI_, _psbE_, _rpl33_, _rps18_, _psbH_, _rrn23_, and _psaC_) were found to be identical, with a nucleotide diversity of zero. The average diversity of the 60 genes was 0.001676, while the average diversity of intergenic spacer regions was 0.00435, approximately 2.6 times that of the genes (Fig 6). Among the intergenic spacer regions, the highest nucleotide diversity was exhibited by _trnH-GUG-psbA_ (0.01458), followed by _rpl32-trnL-UAG_ (0.01225), _psbZ-trnG-GCC_ (0.01148), _trnK-UUU-rps16_ (0.01144), and _ndhF-rpl32_ (0.01112). The alignment lengths of these regions ranged from 366 to 1,829 bp (Fig 6).

## Selective pressure analysis

The ratio of nonsynonymous (Ka) to synonymous (Ks) substitutions was calculated to quantify the evolutionary selective pressure on the _F. sarmentosa_ complex, with _F. simplicis-sima_ used as the reference genome. Overall, Ka/Ks ratios of most genes were < 1 (Fig 7). Additionally, twenty genes were found to contain no substitutions, i.e., both Ka and Ks are zero (shown as "1" in Fig 7). However, five genes (_accD_, _matK_, _ndhF_, _rpoA_, and _ycf1_) had partial Ka/Ks ratios over 1, which are potential signals of positive selection. An infinite Ka/Ks ratio (Ka > 0 and Ks = 0) existed in 27 genes (Fig 7). Notably, ten genes (_accD_, _clpP_,

**Table 2. The relative synonymous codon usage (RSCU) of all 64 codons.** The taxa are represented by the numbers indicated in Table 1.

| Codon | AA | RSCU (relative synonymous codon usage) | | | | | | | | | | | | |
|---|---|---|---|---|---|---|---|---|---|---|---|---|---|---|
| | | 1 | 2 | 3 | 4 | 5 | 6 | 7 | 8 | 9 | 10 | 11 | 12 | 13 |
| UGA | * | 0.662 | 0.662 | 0.662 | 0.662 | 0.662 | 0.662 | 0.662 | 0.662 | 0.662 | 0.662 | 0.662 | 0.662 | 0.662 |
| UAG | * | 0.701 | 0.701 | 0.74 | 0.701 | 0.74 | 0.701 | 0.74 | 0.74 | 0.74 | 0.74 | 0.74 | 0.74 | 0.74 |
| UAA | * | 1.636 | 1.636 | 1.597 | 1.636 | 1.597 | 1.636 | 1.597 | 1.597 | 1.597 | 1.597 | 1.597 | 1.597 | 1.597 |
| GCU | A | 1.853 | 1.853 | 1.853 | 1.853 | 1.862 | 1.853 | 1.854 | 1.861 | 1.863 | 1.869 | 1.861 | 1.862 | 1.857 |
| GCG | A | 0.413 | 0.413 | 0.413 | 0.413 | 0.415 | 0.413 | 0.412 | 0.404 | 0.404 | 0.411 | 0.404 | 0.415 | 0.413 |
| GCC | A | 0.594 | 0.594 | 0.6 | 0.597 | 0.589 | 0.594 | 0.592 | 0.591 | 0.591 | 0.589 | 0.591 | 0.589 | 0.59 |
| GCA | A | 1.14 | 1.14 | 1.135 | 1.136 | 1.134 | 1.14 | 1.142 | 1.144 | 1.143 | 1.132 | 1.144 | 1.134 | 1.141 |
| UGU | C | 1.57 | 1.575 | 1.569 | 1.569 | 1.561 | 1.561 | 1.569 | 1.569 | 1.569 | 1.561 | 1.569 | 1.561 | 1.569 |
| UGC | C | 0.43 | 0.425 | 0.431 | 0.431 | 0.439 | 0.439 | 0.431 | 0.431 | 0.431 | 0.439 | 0.431 | 0.439 | 0.431 |
| GAU | D | 1.596 | 1.596 | 1.596 | 1.596 | 1.596 | 1.598 | 1.594 | 1.595 | 1.595 | 1.597 | 1.595 | 1.598 | 1.593 |
| GAC | D | 0.404 | 0.404 | 0.404 | 0.404 | 0.404 | 0.402 | 0.406 | 0.405 | 0.405 | 0.403 | 0.405 | 0.402 | 0.407 |
| GAG | E | 0.462 | 0.467 | 0.464 | 0.464 | 0.459 | 0.463 | 0.464 | 0.465 | 0.466 | 0.461 | 0.465 | 0.459 | 0.463 |
| GAA | E | 1.538 | 1.533 | 1.536 | 1.536 | 1.541 | 1.537 | 1.536 | 1.535 | 1.534 | 1.539 | 1.535 | 1.541 | 1.537 |
| UUU | F | 1.368 | 1.367 | 1.374 | 1.367 | 1.373 | 1.369 | 1.372 | 1.37 | 1.37 | 1.372 | 1.371 | 1.373 | 1.37 |
| UUC | F | 0.632 | 0.633 | 0.626 | 0.633 | 0.627 | 0.631 | 0.628 | 0.63 | 0.63 | 0.628 | 0.629 | 0.627 | 0.63 |
| GGU | G | 1.34 | 1.34 | 1.338 | 1.338 | 1.341 | 1.334 | 1.342 | 1.337 | 1.338 | 1.342 | 1.337 | 1.341 | 1.342 |
| GGG | G | 0.609 | 0.606 | 0.61 | 0.612 | 0.612 | 0.609 | 0.609 | 0.612 | 0.61 | 0.613 | 0.612 | 0.612 | 0.609 |
| GGC | G | 0.404 | 0.402 | 0.404 | 0.407 | 0.397 | 0.407 | 0.402 | 0.405 | 0.404 | 0.396 | 0.405 | 0.397 | 0.402 |
| GGA | G | 1.647 | 1.652 | 1.648 | 1.643 | 1.651 | 1.649 | 1.647 | 1.647 | 1.648 | 1.649 | 1.647 | 1.651 | 1.647 |
| CAC | H | 0.463 | 0.468 | 0.46 | 0.465 | 0.457 | 0.463 | 0.465 | 0.464 | 0.464 | 0.457 | 0.464 | 0.457 | 0.464 |
| CAU | H | 1.537 | 1.532 | 1.54 | 1.535 | 1.543 | 1.537 | 1.535 | 1.536 | 1.536 | 1.543 | 1.536 | 1.543 | 1.536 |
| AUU | I | 1.471 | 1.471 | 1.471 | 1.477 | 1.47 | 1.473 | 1.471 | 1.468 | 1.467 | 1.47 | 1.468 | 1.47 | 1.47 |
| AUA | I | 0.965 | 0.967 | 0.959 | 0.965 | 0.965 | 0.966 | 0.962 | 0.964 | 0.963 | 0.963 | 0.964 | 0.965 | 0.964 |
| AUC | I | 0.564 | 0.562 | 0.57 | 0.558 | 0.565 | 0.561 | 0.567 | 0.568 | 0.57 | 0.567 | 0.568 | 0.565 | 0.566 |
| AAA | K | 1.531 | 1.534 | 1.534 | 1.533 | 1.528 | 1.53 | 1.534 | 1.536 | 1.537 | 1.527 | 1.536 | 1.528 | 1.539 |
| AAG | K | 0.469 | 0.466 | 0.466 | 0.467 | 0.472 | 0.47 | 0.466 | 0.464 | 0.463 | 0.473 | 0.464 | 0.472 | 0.461 |
| CUA | L | 1.162 | 1.16 | 1.169 | 1.166 | 1.145 | 1.169 | 1.158 | 1.155 | 1.154 | 1.146 | 1.155 | 1.145 | 1.16 |
| CUC | L | 0.521 | 0.522 | 0.528 | 0.518 | 0.52 | 0.521 | 0.518 | 0.516 | 0.516 | 0.52 | 0.516 | 0.52 | 0.515 |
| CUG | L | 0.504 | 0.501 | 0.5 | 0.501 | 0.513 | 0.504 | 0.504 | 0.506 | 0.505 | 0.513 | 0.506 | 0.513 | 0.505 |
| CUU | L | 1.813 | 1.816 | 1.802 | 1.814 | 1.823 | 1.806 | 1.82 | 1.823 | 1.825 | 1.821 | 1.823 | 1.823 | 1.819 |
| UUA | L | 1.249 | 1.251 | 1.25 | 1.249 | 1.254 | 1.249 | 1.25 | 1.254 | 1.254 | 1.251 | 1.254 | 1.252 | 1.25 |
| UUG | L | 0.751 | 0.749 | 0.75 | 0.751 | 0.746 | 0.751 | 0.75 | 0.746 | 0.746 | 0.749 | 0.746 | 0.748 | 0.75 |
| AUG | M | 1 | 1 | 1 | 1 | 1 | 1 | 1 | 1 | 1 | 1 | 1 | 1 | 1 |
| AAC | N | 0.446 | 0.444 | 0.442 | 0.446 | 0.452 | 0.446 | 0.444 | 0.443 | 0.443 | 0.454 | 0.443 | 0.453 | 0.445 |
| AAU | N | 1.554 | 1.556 | 1.558 | 1.554 | 1.548 | 1.554 | 1.556 | 1.557 | 1.557 | 1.546 | 1.557 | 1.547 | 1.555 |
| CCA | P | 1.166 | 1.16 | 1.165 | 1.163 | 1.154 | 1.166 | 1.162 | 1.162 | 1.165 | 1.158 | 1.162 | 1.154 | 1.16 |
| CCC | P | 0.68 | 0.681 | 0.679 | 0.677 | 0.678 | 0.676 | 0.68 | 0.684 | 0.684 | 0.678 | 0.684 | 0.678 | 0.684 |
| CCU | P | 1.592 | 1.596 | 1.586 | 1.598 | 1.596 | 1.597 | 1.592 | 1.592 | 1.591 | 1.596 | 1.592 | 1.596 | 1.591 |
| CCG | P | 0.562 | 0.563 | 0.57 | 0.562 | 0.573 | 0.562 | 0.566 | 0.562 | 0.561 | 0.568 | 0.562 | 0.573 | 0.565 |
| CAA | Q | 1.585 | 1.588 | 1.58 | 1.586 | 1.587 | 1.585 | 1.582 | 1.585 | 1.583 | 1.584 | 1.585 | 1.587 | 1.582 |
| CAG | Q | 0.415 | 0.412 | 0.42 | 0.414 | 0.413 | 0.415 | 0.418 | 0.415 | 0.417 | 0.416 | 0.415 | 0.413 | 0.418 |
| AGA | R | 1.496 | 1.5 | 1.5 | 1.497 | 1.502 | 1.494 | 1.496 | 1.499 | 1.501 | 1.502 | 1.499 | 1.502 | 1.497 |
| AGG | R | 0.504 | 0.5 | 0.5 | 0.503 | 0.498 | 0.506 | 0.504 | 0.501 | 0.499 | 0.498 | 0.501 | 0.498 | 0.503 |
| CGA | R | 1.6 | 1.596 | 1.604 | 1.596 | 1.602 | 1.601 | 1.596 | 1.591 | 1.594 | 1.602 | 1.591 | 1.602 | 1.596 |
| CGC | R | 0.405 | 0.399 | 0.398 | 0.399 | 0.403 | 0.399 | 0.399 | 0.399 | 0.398 | 0.403 | 0.399 | 0.403 | 0.399 |
| CGG | R | 0.426 | 0.425 | 0.419 | 0.425 | 0.423 | 0.43 | 0.414 | 0.419 | 0.419 | 0.423 | 0.419 | 0.423 | 0.414 |

*(Continued)*

**Table 2.** (Continued)

| Codon | AA | RSCU (relative synonymous codon usage) | | | | | | | | | | | | |
|-------|-----|-------|-------|-------|-------|-------|-------|-------|-------|-------|-------|-------|-------|-------|
| | | 1 | 2 | 3 | 4 | 5 | 6 | 7 | 8 | 9 | 10 | 11 | 12 | 13 |
| CGU | R | 1.569 | 1.581 | 1.579 | 1.581 | 1.571 | 1.57 | 1.591 | 1.591 | 1.589 | 1.571 | 1.591 | 1.571 | 1.591 |
| AGC | S | 0.447 | 0.456 | 0.455 | 0.46 | 0.451 | 0.45 | 0.454 | 0.454 | 0.454 | 0.45 | 0.454 | 0.447 | 0.454 |
| AGU | S | 1.553 | 1.544 | 1.545 | 1.54 | 1.549 | 1.55 | 1.546 | 1.546 | 1.546 | 1.55 | 1.546 | 1.553 | 1.546 |
| UCA | S | 1.111 | 1.111 | 1.115 | 1.114 | 1.116 | 1.107 | 1.111 | 1.109 | 1.108 | 1.116 | 1.11 | 1.116 | 1.109 |
| UCC | S | 0.817 | 0.817 | 0.809 | 0.817 | 0.803 | 0.814 | 0.806 | 0.812 | 0.816 | 0.8 | 0.813 | 0.803 | 0.808 |
| UCG | S | 0.449 | 0.449 | 0.448 | 0.452 | 0.448 | 0.452 | 0.45 | 0.448 | 0.447 | 0.448 | 0.448 | 0.448 | 0.446 |
| UCU | S | 1.624 | 1.622 | 1.628 | 1.617 | 1.632 | 1.627 | 1.632 | 1.631 | 1.629 | 1.635 | 1.629 | 1.632 | 1.636 |
| ACC | T | 0.701 | 0.699 | 0.701 | 0.701 | 0.701 | 0.701 | 0.702 | 0.695 | 0.697 | 0.702 | 0.695 | 0.701 | 0.699 |
| ACA | T | 1.187 | 1.189 | 1.182 | 1.185 | 1.177 | 1.185 | 1.176 | 1.177 | 1.179 | 1.179 | 1.177 | 1.177 | 1.174 |
| ACG | T | 0.417 | 0.416 | 0.422 | 0.417 | 0.417 | 0.417 | 0.422 | 0.426 | 0.425 | 0.418 | 0.426 | 0.417 | 0.426 |
| ACU | T | 1.695 | 1.696 | 1.695 | 1.697 | 1.706 | 1.697 | 1.7 | 1.701 | 1.698 | 1.702 | 1.701 | 1.706 | 1.701 |
| GUU | V | 1.463 | 1.462 | 1.461 | 1.468 | 1.466 | 1.466 | 1.461 | 1.458 | 1.461 | 1.46 | 1.458 | 1.466 | 1.461 |
| GUG | V | 0.513 | 0.511 | 0.522 | 0.513 | 0.513 | 0.51 | 0.513 | 0.515 | 0.521 | 0.516 | 0.515 | 0.513 | 0.513 |
| GUC | V | 0.405 | 0.406 | 0.405 | 0.403 | 0.409 | 0.405 | 0.406 | 0.411 | 0.407 | 0.415 | 0.411 | 0.409 | 0.406 |
| GUA | V | 1.618 | 1.621 | 1.613 | 1.616 | 1.612 | 1.618 | 1.62 | 1.616 | 1.612 | 1.609 | 1.616 | 1.612 | 1.62 |
| UGG | W | 1 | 1 | 1 | 1 | 1 | 1 | 1 | 1 | 1 | 1 | 1 | 1 | 1 |
| UAC | Y | 0.365 | 0.364 | 0.364 | 0.365 | 0.364 | 0.361 | 0.366 | 0.366 | 0.365 | 0.364 | 0.366 | 0.364 | 0.363 |
| UAU | Y | 1.635 | 1.636 | 1.636 | 1.635 | 1.636 | 1.639 | 1.634 | 1.634 | 1.635 | 1.636 | 1.634 | 1.636 | 1.637 |

*ndhK*, *rbcL*, *rpl20*, *rpl22*, *rpl23*, *rpoC1*, *rps15*, and *rps4*) possessed infinite Ka/Ks ratios in more than half of the taxa.

## Phylogenetic analysis

The chloroplast phylogenomic ML tree illustrated a well-supported phylogenetic relationship in the genus *Ficus* (Fig 8). Subgenus *Pharmacosycea* sect. *Pharmacosycea* was strongly supported as a sister group to the rest of the genus *Ficus* (SH-aLRT = 100 and MLBS = 100) Additionally, a clade including subg. *Urostigma* sect. *Galoglychia* and *Americana*, subg. *Sycomorus* sect. *Sycomorus*, and a few species of subg. *Pharmacosycea* sect. *Oreosycea* (subser. *Ablbipilae* in Corner's system) (clade A) was found to be sister to the remainder of *Ficus* except sect. *Pharmacosycea*. Aside from these two clades, the remaining *Ficus* taxa formed three clades (clades B, C, and D in Fig 8), with unstable support (SH-aLRT = 64.9 and MLBS = 73). Clade B (SH-aLRT = 99.9 and MLBS = 100) was comprised of five different subgenera, while clade C (SH-aLRT = 100 and MLBS = 100) included all species of subg. *Sycidium* as well as members of each of the other five subgenera. Clade D (SH-aLRT = 100 and MLBS = 100) contained only two species in subg. *Pharmacosycea* sect. *Oreosycea*. On the whole, all six subgenera were found not to be monophyletic.

With the inclusion of eleven additional samples, a phylogenomic analysis was carried out on 24 samples representing the 13 taxa in the *F. sarmentosa* complex (Fig 8). The results showed that the *F. sarmentosa* complex failed to form a monophyletic group. Except three samples which are unexpectedly embedded in a distinct clade with other members of subg. *Synoecia* (clade E), two distinct lineages were recognized, with the majority of nodes within these two lineages being well-supported (clades F and G). The six *F. sarmentosa* varieties were found to be scattered across clades E, F, and G, and only two individuals of *F. sarmentosa* var. *lacrymans* clustered together. The six *F. pumila* samples were also found not to be a monophyletic group embedded by *F. sarmentosa* var. *thunbergii*.

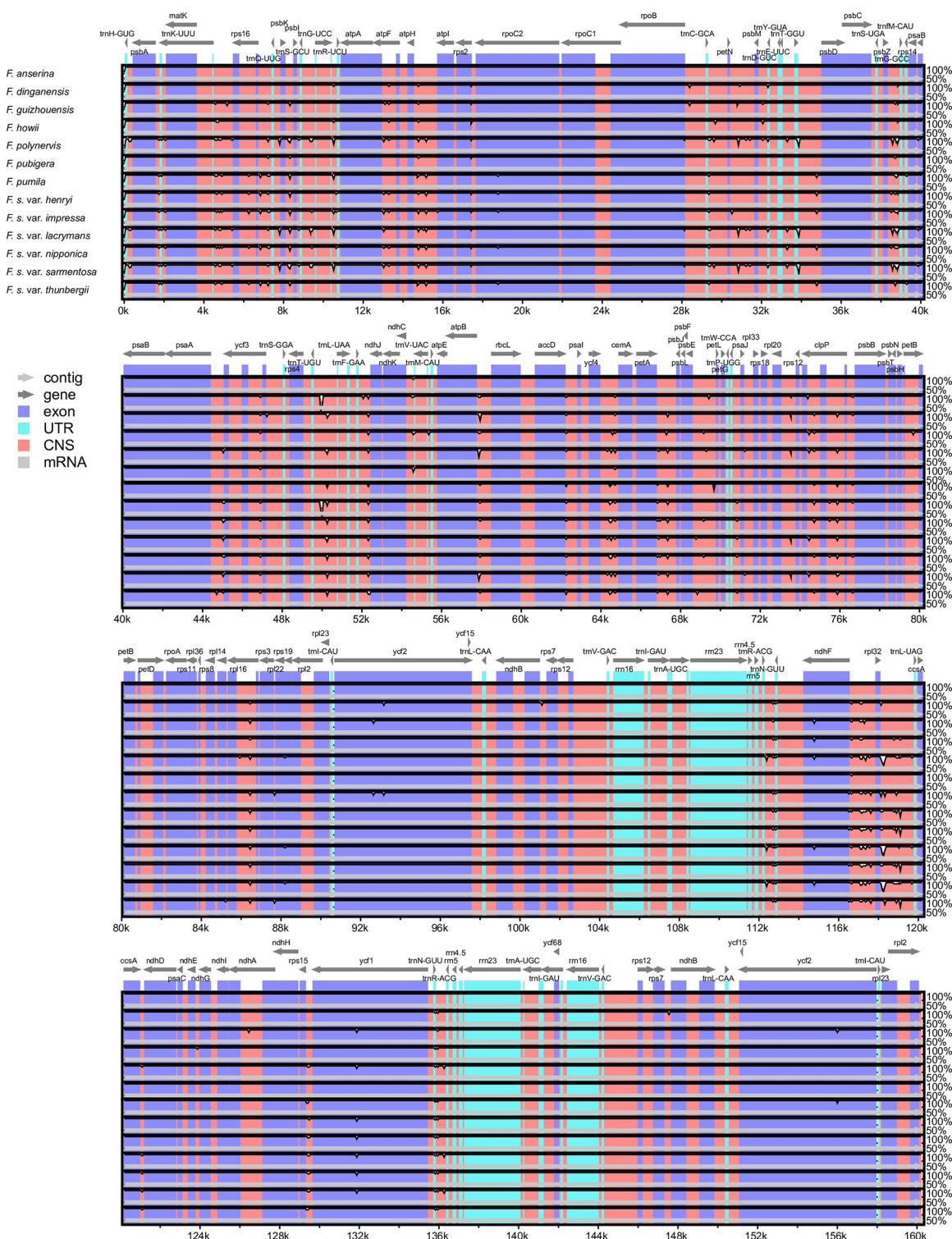

**Fig 5. Comparison of complete chloroplast genomes among 13 taxa in the *F. sarmentosa* complex with *F. anserina* as a reference.**
Thick, gray arrows above the alignment indicate the orientation and position of each gene. A cut-off of 70% identity was chosen for the plots. The Y-axis represents the identity percentage, ranging from 50 to 100%.

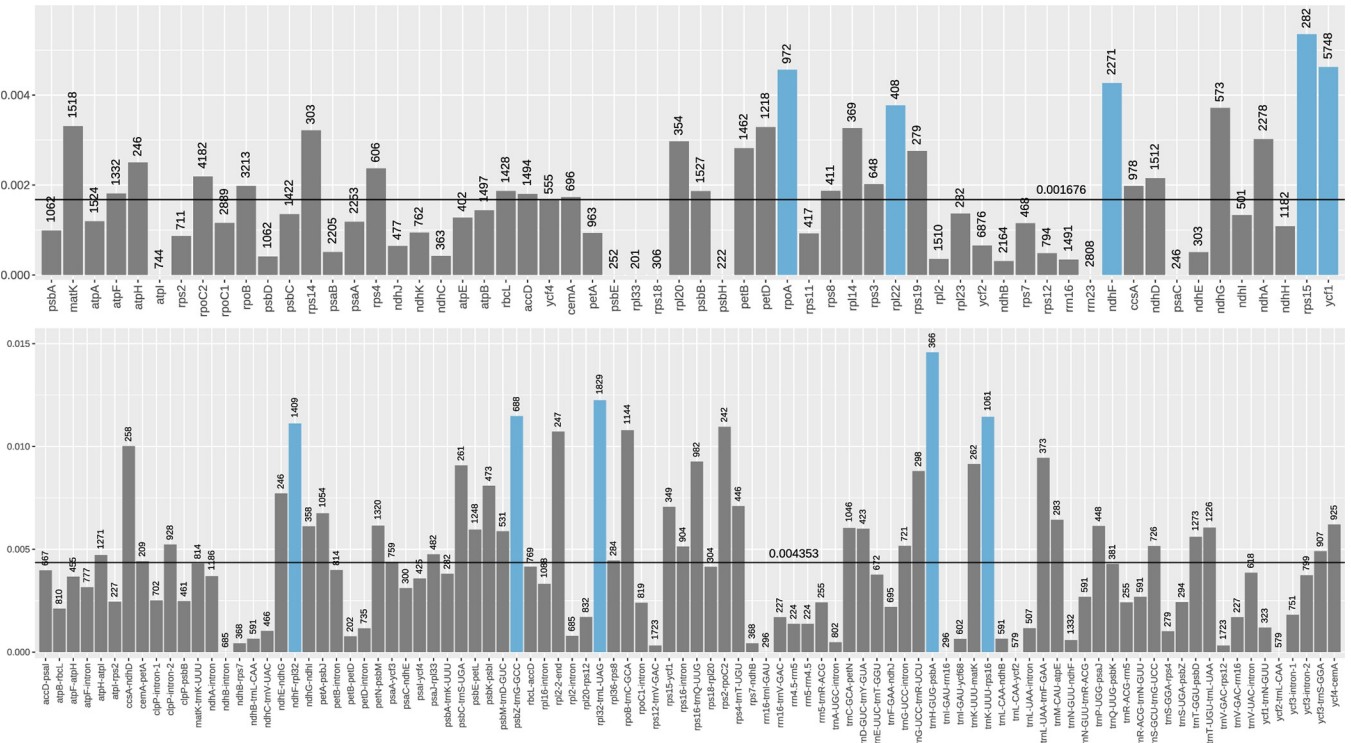

**Fig 6. Nucleotide diversity of genes and intergenic spacer regions among 13 taxa in the *F. sarmentosa* complex.** The alignment lengths are indicated on the bars. The horizontal lines indicate the average nucleotide diversity of genes and intergenic spacer regions, respectively. The top five genes or intergenic spacers with the highest nucleotide diversity are highlighted in blue.

## Discussion

### The differentiation and diversity of the chloroplast genome in the *F. sarmentosa* complex

To date, vast comparative chloroplast genomic studies have been conducted in a wide range of taxonomic levels, such as order (such as Dipsacales [56] and Saxifragales [57]), family (such as

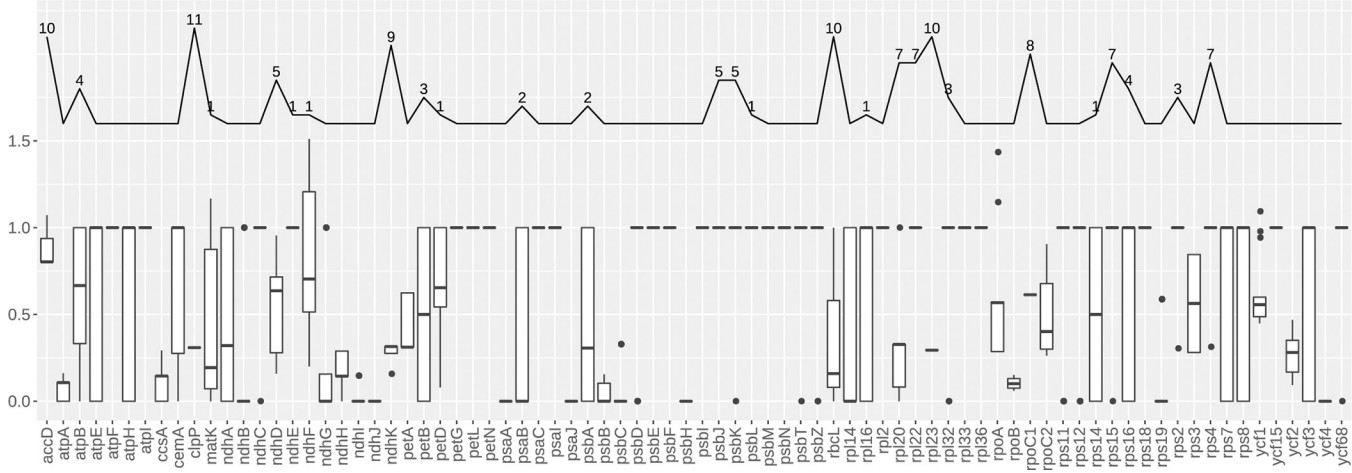

**Fig 7. Boxplot of Ka/Ks ratios for 80 unique CDS regions.** The value 1.0 represents the situation where both Ka and Ks equal zero. The line chart superimposed upon the boxplot demonstrates the frequency of infinite Ka/Ks ratios (Ka > 0 and Ks = 0), with detailed numbers labeled simultaneously.

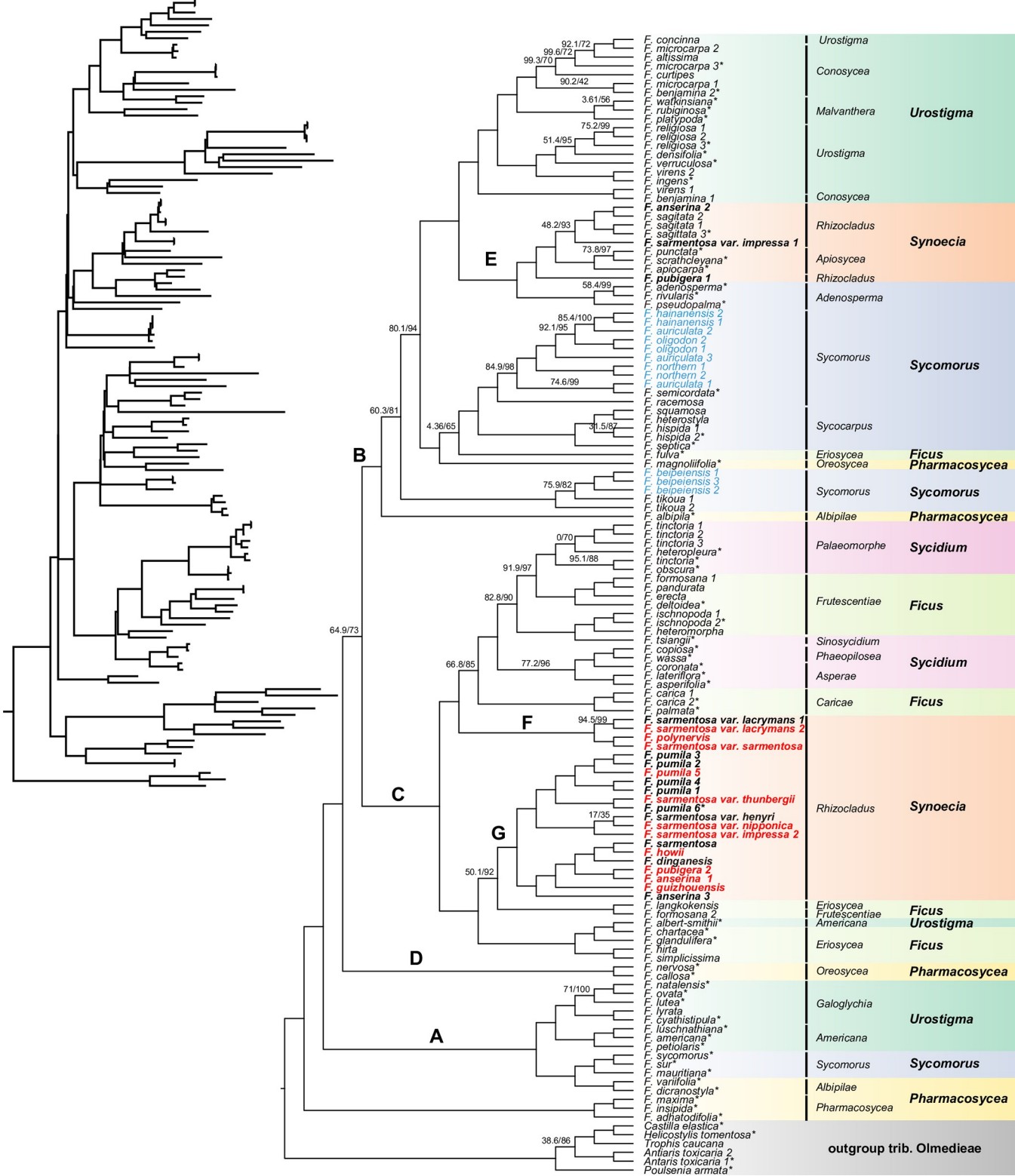

**Fig 8. The maximum likelihood (ML) phylogenetic tree of 123 chloroplast genomes in *Ficus* with six Olmedieae genomes as the outgroup.** Only the branches with either SH-aLRT or ultrafast bootstrap < 95% were annotated by corresponding values. The starred tip names indicate genomes obtained from Bruun-Lund et al. [26]; the red names indicate genomes obtained in this study; the blue names indicate members of the *F. auriculata* complex; and the bold names indicate members of the *F. sarmentosa complex*. The subgenus and section division of *Ficus* are annotated to the right of tip names. The topology of the ML tree is shown in the upper left corner (excluding the outgroup).

Orchidaceae [58] and Zingiberaceae [59]), and genus (such as *Camellia* [60], *Lindera* [61], *Gossypium* [62], and *Ficus* [17, 51, 63]). However, less research has focused on the comparative genomics of taxa undergoing recent speciation, such as the species complex [29]. Although the structural conservation of the chloroplast genome at low taxonomic levels is well-characterized [64–67], the detailed patterns of genomic differentiation and diversity among closely related species remain largely unknown. Therefore, comprehensive comparisons between closely related species are necessary to improve our understanding of the mechanisms, rates, or directionality of genome evolution during the early stages after speciation [29].

In this study, we investigated the evolutionary dynamics of thirteen high-quality chloroplast genomes from the *F. sarmentosa* complex. Overall, the lengths of both whole-genome and quadripartite regions were quite similar among taxa, with only 0.2288%, 0.3784%, 0.2188%, and 0.1236% variation among whole genomes, LSCs, SSCs, and IRs, respectively (Table 1, Fig 1). Furthermore, the number, content, and orientation of annotated genes among all 13 genomes were identical. The IR boundaries (JLB, JSB, JSA, and JLA) were also relatively coincident among the 13 genomes, being located at the same loci with only slight variation in the distance to the starting or ending bases. For long repeat sequences, all four repeat units were shared among all 13 plastomes, and the number of repeat units and their proportions exhibited only slight differentiation. For example, the proportion of palindromic sequences ranged from 45.16 to 61.54% (Fig 3). Similarly, in SSR regions, the proportion of the mononucleotide repeat units ranged from 72.63 to 85.96% (Fig 4).

The high conservation exhibited across the *F. sarmentosa* complex is consistent with other studies on closely related taxa. For example, a study of 22 closely related *Oryza* species indicated that conservation was common at lower taxonomic levels [29]. Even in morphologically diverse shrub willows (*Salix*), such high conservation is still exhibited [68]. A study of four peanut varieties serves as a more extreme example, reporting perfectly identical IR boundary junction positions [69]. Although comparative chloroplast genomic studies have rarely focused on closely related species, the high conservation of chloroplast genome among closely related species is recognized based on our work and other related research.

## Chloroplast genomic evolutionary hotspots

Although chloroplast genomes exhibit high conservation among closely related species, discrepancy and heterogeneity have also been widely observed across the whole genomes. Overall, the variable sites of the two single-copy regions (LSC and SSC) are more abundant than IR regions in both the genus *Ficus* (Fig 5) and most other plant groups [56, 70, 71]. Chloroplast regions with different mutation rates are appropriate for a range of evolutionary research. In general, conservative coding genes are suitable for deep phylogenetic inferences at the family-level [28, 72, 73] or higher [74–76], whereas highly variable regions are appropriate for studies of biogeography, species delimitation, population genetics, and phylogenetic reconstruction at lower infra-generic levels [77, 78].

To date, more than 20 regions have been recommended as alternative loci for phylogenetics, species delimitation, and barcoding, including *matK*, *rbcL*, *trnH-psbA*, *ycf1*, *ycf1-ndhF*, among others [78–80]. However, these loci had few informative sites in the *F. sarmentosa* complex (Fig 6), suggesting that evolutionary heterogeneity of chloroplast loci may be relatively common among different plant groups. In this study, we mined five hyper-variable intergenic regions at the level of the species complex, i.e., *trnH-GUG-psbA*, *rpl32-trnL-UAG*, *psbZ-trnG-GCC*, *trnK-UUU-rps16* and *ndhF-rpl32*. However, other studies of comparative chloroplast genomics in *Ficus* identified entirely different hyper-variable intergenic regions. For example, Xia et al. [63] identified *trnS-GCU-trnG-UCC*, *trnT-GGU-psbD*, *trnV-UAC-trnM-CAU*, *clpP-*

*psbB*, *ndhF-trnL-UAG*, *trnL-UAG-ccsA*, *ndhD-psaC*, and *ycf1*, and Zhang et al. [17] identified t*rnL-UAG-rpl32*, *trnE-UUC-psbD*, *trnK-UUU-rps16*, *rpoB-trnC-GCA*, and *petN-psbM*. These disparate results suggest that the evolutionary dynamics of chloroplast genomes varies across both groups and taxonomic ranks. Therefore, we suggest that chloroplast loci should be chosen cautiously according to the research objective, plant group, and taxonomic level.

### The plastome phylogeny of the genus *Ficus* and the phylogenetic performance in closely related species

Based on a compilation of nearly all available *Ficus* chloroplast genomes, we obtained a robust ML phylogenetic tree with the majority of nodes exhibiting high bootstrap and SH-aLRT values, particularly the deep nodes (Fig 8). Overall, our ML phylogenetic tree is largely consistent with previous research [17, 26], including the systematic position of subg. *Pharmacosycea* sect. *Pharmacosycea* and the mysterious displacement of certain individuals (*F. fulva*, *F. magnoliifolia*, *F. albipila*, *F. albert-smithii*, and *F. pumila*). However, analysis of our extended dataset highlighted an increase in non-monophyletic groups, such as subg. *Urostigma* sect. *Urostigma* and *Conosycea* (Fig 8). Notably, there were numerous incongruences between the chloroplast cladogram and previously published nuclear trees. For example, the chloroplast-based tree failed to support either *Ficus* subg. *Sycomorus* or subg. *Sycidium* as monophyletic groups, whereas these groups are well-confirmed in nuclear phylogenies [3, 12–14, 81]. Subg. *Synoecia* was also divided into three different clades (Fig 8). Further research into the displaced species, such as *F. fulva*, *F. ablert-smithii*, *F. magnoliilolia*, and *F. albipila*, may reveal host shifts or nonspecific pollination between fig trees and fig wasps [26]. Considering that a stable and comprehensively-sampled nuclear phylogenomic framework for the genus *Ficus* is still lacking, precise identification of these hybridization events will require more robust nuclear genome data as well as data from the associated fig wasps.

Compared to the nuclear phylogeny [4], we discovered a disparate evolutionary relationship among taxa in the *F. sarmentosa* complex, including three distinct clades (Fig 8, clades E, F, and G). Three samples from the complex dispersed into the clade E mixing with *F. sagittata* and the members of subg. *Synoecia* sect. *Apiosycea*. Unless more data support hybridization, misidentification may be an alternative explanation. Neither geographic nor morphological traits could be detected to support the split of clades F and G. Hybridization between *F. pumila* and *F. sarmentosa* var. *thunbergii* might exist, considering that the latter embedded into the former within clade G. Although our current samples are insufficient to fully resolve the taxonomy of the *F. sarmentosa* complex, the phylogenetic resolution provided by chloroplast genomes in the complex appears to be promising, as almost all the nodes are strongly supported. The chloroplast genome has also shown high discriminability across the *F. auriculata* complex (Fig 8, blue labels) [82–84]. The relationships between four taxa (*F. auriculata*, *F. oligodon*, *F. hainanensis*, and *F. beipeiensis*) in the complex were well-resolved, while *F. beipeiensis* shared a distinct phylogenetic relationship with the climbing fig tree *F. tikoua*. Moreover, a new linage (*F. northern*) was identified based on chloroplast genomics, suggesting a promising segue into further exploration of the cryptic species (Fig 8, blue labels) [17]. Chloroplast genomes have been used to reconstruct high-resolution phylogenetic trees in other closely related species groups, such as peanut [85], rice [29], willow [68], and orchardgrass [86]. Taken together, the chloroplast genome appears to be a promising tool for exploring the evolutionary relationships between closely related species and even species complexes, although cyto-nuclear discordances often exist.

## Conclusions

In this study, eleven *F. sarmentosa* complex chloroplast genomes were newly sequenced and characterized. Sequence lengths, IR boundaries, repeat sequences, and codon usage were

compared among these eleven, and two previously-reported, chloroplast genomes, indicating that these parameters were highly conserved across taxa. However, heterogeneity was found in both nucleotide diversity and selective pressure among segments. We characterized ten evolutionary hotspot regions (*rps15*, *ycf1*, *rpoA*, *ndhF*, *rpl22*, *trnH*-GUG-*psbA*, *rpl32-trnL-UAG*, *psbZ-trnG-GCC*, *trnK-UUU-rps*16 and *ndhF-rpl32*). Phylogenomic analysis indicated that chloroplast genomes show promise for inferring the phylogenetic relationships between closely related groups, despite cyto-nuclear discordance.

## Supporting information

**S1 Table. The collecting information of samples in the study.**
(XLSX)

**S2 Table. Detailed information for the high-throughput sequencing data in the study.**
(XLSX)

## Acknowledgments

We are grateful to Dr. Hong-Qing Li at East China Normal University, Dr. Zhi-Hui Su and Ms. Sasaki Ayako at Osaka University, Dr. Yong Chen and Dr. Kai-Liang Liu at Ningde Normal University, and Mr. Zhen Liu at Forestry and Grassland Administration of Motuo County for their support in wild collecting. The authors would like to thank TopEdit (www.topeditsci.com) for its linguistic assistance during the preparation of this manuscript.

## Author Contributions

**Conceptualization:** De-Shun Zhang, Chi-Yuan Yao.

**Data curation:** Lu Zou.

**Formal analysis:** Zhen Zhang.

**Funding acquisition:** Chi-Yuan Yao.

**Investigation:** Zhen Zhang.

**Methodology:** Zhen Zhang.

**Project administration:** Chi-Yuan Yao.

**Resources:** Chi-Yuan Yao.

**Software:** Zhen Zhang.

**Visualization:** De-Shun Zhang.

**Writing – original draft:** Zhen Zhang, Lu Zou.

**Writing – review & editing:** Zhen Zhang.

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
