## [Decision Letter · Decision Letter 0]

15 Sep 2022

PONE-D-22-13870The comparison of chloroplast genomes and phylogenomics at closely related species: a case of Ficus sarmentosa complex (Moraceae)PLOS ONE

Dear Dr. Yao,

Thank you for submitting your manuscript to PLOS ONE. After careful consideration, we feel that it has merit but does not meet PLOS ONE’s publication criteria as it currently stands. Therefore, we invite you to submit a revised version of the manuscript that addresses the points raised during the review process.

 Reviewer 1 requests to regard more new literature and to discuss better the phylogenetic results.Reviewer 2 is much more critical and requests to frame better the aims of the study. While there are many valuable details on the chloroplast genomes there are insufficient comparisons with previous phylogenetic work. Clarify all sources of data and explain why you regard previous cp genomes of low quality. The speculations on cyto-nuclear conflict should be left out as this was not studied. Introduction and discussion should be re-written accordingly.From the editor's view I would like to add that for discussion also potential homoplasy in chloroplast genomes should be considered (see eg. Wagner et al. 2021 doi: 10.3389/fpls.2021.662715 on Salix).The language needs attention, please ask a native English speaker for help.

We look forward to receiving your revised manuscript. Since a major revision was indicated the revised version will undergo another review process.

Kind regards,

Elvira Hörandl

Academic Editor

PLOS ONE

Journal Requirements:

2. Please include your tables as part of your main manuscript and remove the individual files. Please note that supplementary tables (should remain/ be uploaded) as separate "supporting information" files

“This research was funded by National Natural Science Foundation of China from Dr. De-Shun Zhang, grant number 32071824. The funder De-Shun Zhang implemented the conceptualization and visualization of the study.”

“NO authors have competing interests”

Reviewers' comments:

Reviewer's Responses to Questions

**Comments to the Author**

1. Is the manuscript technically sound, and do the data support the conclusions?

Reviewer #1: Yes

Reviewer #2: Partly

2. Has the statistical analysis been performed appropriately and rigorously? 

Reviewer #1: Yes

Reviewer #2: I Don't Know

3. Have the authors made all data underlying the findings in their manuscript fully available?

Reviewer #1: No

Reviewer #2: Yes

4. Is the manuscript presented in an intelligible fashion and written in standard English?

Reviewer #1: No

Reviewer #2: No

5. Review Comments to the Author

Reviewer #1: The manuscript entitled "The comparison of chloroplast genomes and phylogenomics at closely related species: a case of Ficus sarmentosa complex (Moraceae)" by Zhang et al. tried to estimate phylogenetic relationships of a species complex of the genus Ficus using newly obtained 13 chrloroplast sequences mined from whole genome sequencing reads. The manuscript are well-written, and I think it seems be satisfying most of publication criteria of this journal but I have some concerns as follows.

Major comments:

1. The taxonomic groups used by the authors will be not used by some other authors as in table 1 of Rasplus et al. (2021), such as the subsection class. So it will be better to provide the information on which classification followed by the authors as the table 1 of Rasplus et al. (2021).

Rasplus, J. Y., Rodriguez, L. J., Sauné, L., Peng, Y. Q., Bain, A., Kjellberg, F., ... & Cruaud, A. (2021). Exploring systematic biases, rooting methods and morphological evidence to unravel the evolutionary history of the genus Ficus (Moraceae). Cladistics, 37(4), 402-422.

2. Although the authors did not mention, all subgenera seems paraphyletic in the phylogeny. The authors should discuss this result itself and the utility of chloroplast genome to estimate phylogeny of the genus Ficus. And lack of the section names in each clade of fig 8, some sentences especially results of phylogenetic analysis are difficult to follow.

Minor comments:

Line 2 and elsewhere: Ficus sarmentosa complex -> the Ficus sarmentosa complex

Lines 15-16: The authors seem used more numbers of sequences in the phylogenetic analysis than two, please rephrase to make clear how many sequences obtained from the previous studies and which analysis used which sequences.

Lines 42-44: "the latest research": the following research seems more recent than that the authors cite, please refer to this research also, if it related to this sentences.

Rasplus, J. Y., Rodriguez, L. J., Sauné, L., Peng, Y. Q., Bain, A., Kjellberg, F., ... & Cruaud, A. (2021). Exploring systematic biases, rooting methods and morphological evidence to unravel the evolutionary history of the genus Ficus (Moraceae). Cladistics, 37(4), 402-422.

Lines 56 and elsewhere: "What’s more", although I'm not native speaker of English, but this phrase may seem too colloquial.

Line 61: "complete genome": please make clear which of complete nuclear genome or chloroplast genome is meant by this phrase.

Lines 62-63: Please cite the reference for Plagiostigma because some authors do not use this taxonomic grouping, such as Cruaud et al. 2012.

Cruaud, A., Rønsted, N., Chantarasuwan, B., Chou, L. S., Clement, W. L., Couloux, A., ... & Savolainen, V. (2012). An extreme case of plant–insect codiversification: figs and fig-pollinating wasps. Systematic Biology, 61(6), 1029-1047.

Lines 106-107: "IR, LSC, SSC": Please spell out these abbreviations in the first use.

Lines 142-144: Please provide the accession numbers of these 126 sequences such as in the supplemental table.

Lines 142-144: "these genomes are unavailable in GenBank currently)": if so, how did the authors obtained those sequences?

Line 144: "138": this may be typo but 126 and 13 sequences will be 139. Please explain what was happen if the number of sequences truly is 138.

Lines 151: Not only the reads of chloroplast but also the basic information on the raw reads containing nuclear reads, such as total reads and percentage of chloroplast reads in the total reads will be informative.

Lines 170-171: Please add the citation for the statement, "infA gene is lost in Ficus".

Lines 254-274: Although the authors did not mention, all subgenera seems paraphyletic in the phylogeny. Please see the major comment.

Line 258 and elsewhere: Ficus -> the genus Ficus

Lines 260 and 261: "basal group": it is better to avoid using this phrase according to the following article.

Krell, F. T., & Cranston, P. S. (2004). Which side of the tree is more basal?. Systematic Entomology, 29(3), 279-281.

Fig. 8: Please add the section name to each clade, some sentences especially results of phylogenetic analysis are difficult to follow because of lack of this information. Please see the major comment.

Table 1: If these accession numbers are only for the chloroplast sequence, please provide the accession numbers for the raw reads from NGS, too.

Reviewer #2: The title and objective do not match the described study with respect to the sampling and discussion. The authors should revisit their objectives or sampling to ensure the manuscript is well-aligned from introducing the background and scope to the chosen sampling and the presentation and discussion of the results.

While the 11 new chloroplast genomes are a valuable contribution and a lot of detailed analysis of the genomes are provided and is clearly the authors strength, unfortunately with the chosen sampling strategy, the study adds very little to its objective of understanding the usefulness of chloroplast genomes to resolve species complexes or the taxonomic problems of the targeted species complex.

Introduction

l.52. The authors state that only a few chloroplast genomes have been reported in Ficus, but in the next line refer to a study by Bruun-Lund et al. (2017) including 59 species of Ficus, which I would not call only a few genomes.

The authors of the submitted manuscript rightly explain that the results from Bruun-Lund showed conflicts between the new plastome topology and previous nuclear studies for both individual species as well as relationships among some sections at deeper levels.

l. 56. However, the authors of the present manuscript appear to ignore this issue arguing "that the majority of genome sequences of Bruun-Lund et al. were relatively incomplete and perhaps disturbed the accuracy of phylogenetic tree due to an abundant of missing data". However, it is not explained what relatively incomplete is and no examples or argumentation is provided for this potentially being an issue.

l. 61. The authors state that "The species of subsect. Plagiostigma is under taxonomically unresolved and genetically unclear, formed a species complex and an ideal candidate for the researches of hybridization, radiation speciation, biogeography, and interaction between plant and pollinator". In addition to a need for revisiting this sentence for unclear English and meaning, the authors should provide a short discussion of the taxonomic history of the F. sarmentosa complex if this is the primary focus of the manuscript. What was learned from the study by Zhang et al. (reference 3, a well-written manuscript by several of the same authors) compared to Berg and Corner (2005).

l. 65. The F. sarmentosa complex or subsect. Plagiostima sensu Berg and Corner (2005) includes at least 10 species and is part of larger taxonomic problem which at the moment includes all the root climbers of subg. Synoecia and it would have been logical and much more interesting to expand on the sampling of this entire subgenus.

If the aim is indeed to resolve the F. sarmentosa complex and discuss the usefulness of chloroplast genomes to resolve closely related species, it would be more appropriate to expand the sampling of this complex with several samples per taxon.

It is also not clear why sampling extends to 61 additional chloroplast genomes of Moraceae. It would appear to be enough to include the Olmedieae based on previous studies.

l. 77. Following the objectives, the authors state "we sampled the plastomes of eleven taxa and then finished the comparative genomics and phylogenomic analysis of F. sarmentosa complex", which does not adequately reflect the results obtainable with this sampling or the discussion of these results.

Materials and methods.

l. 82 onwards. Here you should refer back to the introduction outline of the F. sarmentosa complex and explain your sampling strategy and how well it represents the diversity of your study group.

l. 143. It is stated that: "Thirteen chloroplast genomes, additional 61 available Moraceae chloroplast genomes in GenBank and 65 genomes from the work of Bruun-Lund et al [22] (these genomes are unavailable in GenBank

144 currently) were chosen". Do you mean these genomes are available? It is not clear since they were obviously included in your study.

Results.

Considering your hypothesis is the introduction "that the majority of genome sequences of Bruun-Lund et al. were relatively incomplete and perhaps disturbed the accuracy of phylogenetic tree due to an abundant of missing data", you need to provide a detailed comparison of whether and how your results have indeed improved on completeness and whether this has significantly improved relationships.

This would also provide the necessary background for the discussion l. 332 stating "However, in our focused group Ficus, 35 newly added genomes didn’t affect the linage relationship compared to the previous study of Bruun-lund et al."

l. 259. How does your topology compare to the one published by Bruun-Lund et al. and the taxonomic framework provided by Clement et al. (2020)?

l. 268. "For the 13 taxa in F. sarmentosa complex, two obvious linages were recognized and the most of nodes within these two linages were well-supported (Figure 8)". Please discuss how the findings match previous taxonomic understanding of the complex and in particular clarify that two of the F. sarmentosa samples result in polyphyly of Synoecia.

Discussion.

L. 278-300. If it has been widely demonstrated that the chloroplast genome is relatively conserved and provide little information among closely related species, it is not clear why the authors would focus on exploring this in close relatives or a megadiverse genus like Ficus.

l. 328. The authors state that "The ML phylogenetic tree is generally consistent to previous researches based on hundreds of nuclear genes [63], such as Morus and Streblus are reconfirmed as non-monophyletic groups. The

consistent results revealed the great potential of chloroplast genome in phylogenomics of Moraceae". It is not clear what the point is and why re-analysing previously published data would not be expected to result in more or less the same topology for Moraceae. If you want to discuss the great potential of chloroplast genome in phylogenomics of Moraceae, you could focus on discussing stability of the results among previous studies of which there are quite a few.

l. 340. Please revisit the manuscript including the discussion for sentence clarity possibly with the help of a professional English speaker or consultant. See for example the sentences: "Whereas, current phylogenetic frameworks of Ficus were acquired by only up to six nuclear loci [12, 64] or limited and disproportional samples [14]. So, it is still deficient to distinguish the origin of the cyto-nuclear conflict accurately, such as gene flow (hybridization or introgression) or incomplete linage sorting".

l.343. Again, it should be discussed what your results learns us about the F. sarmentosa complex.

L. 351. Several times it is mentioned that the chloroplast genome helps to identify potential cyto-nuclear discordance, but it is not really discussed how this could be further explored and verified.

l. 365. The conclusion: "Phylogenomic construction indicated that chloroplast genomes show a promising resolution for phylogenetic relationship of closely related group with the evident cyto-nuclear discordance" is not well-explained and is an over-statement and contra=diction of what is actually analysed and the results obtained in this study.

6. PLOS authors have the option to publish the peer review history of their article (what does this mean?). If published, this will include your full peer review and any attached files.

Reviewer #1: No

Reviewer #2: No

---

## [Author Response · Author response to Decision Letter 0]

10 Nov 2022

The responce to the comments from reviewers and editor can be found in the responce letter.

---

## [Editor Report · Decision Letter 1]

14 Nov 2022

PONE-D-22-13870R1The comparison of chloroplast genomes and phylogenomics in the Ficus sarmentosa complex (Moraceae)PLOS ONE

Dear Dr. Yao,

Thank you for submitting your manuscript to PLOS ONE. After careful consideration, we feel that that the revision was sufficient regarding the content, but the language still needs to be improved. Specifically the newly inserted text has flaws in grammar and wording. Please ask a native English speaker for help or use a professional editing service,

We look forward to receiving your revised manuscript.

Kind regards,

Elvira Hörandl

Academic Editor

PLOS ONE
---

## [Editor Report · Decision Letter 2]

13 Dec 2022

PONE-D-22-13870R2Comparison of chloroplast genomes and phylogenomics in the Ficus sarmentosa complex (Moraceae)PLOS ONE

Dear Dr. Yao,

Thank you for submitting your manuscript to PLOS ONE. The manuscript has much improved, but I found still a few linguistic flaws (see in the file PONE-D-22-13870_R2_edit, yellow bubbles in the clean version). After these final corrections it will be accepted.

We look forward to receiving your revised manuscript.

Kind regards,

Elvira Hörandl

Academic Editor

PLOS ONE
---

## [Author Response · Author response to Decision Letter 2]

14 Dec 2022

The language mistakes pointed out by the editor have been simply accepted, so only one key comment was detailedly responded to here.

Response to comments of the editor:

For example, a study of 22 closely related Oryza species indicated that conservation was common at lower taxonomic levels [29]. A study of four peanut varieties serves as a more extreme example, reporting perfectly identical IR boundary junction positions [69]. Although comparative chloroplast genomic studies have rarely focused on closely related species, the high conservation of chloroplast genome among closely related species is recognized based on our work and other related research.

Answer: Done, the study is very relevant, and we have added it. Please see Lines 376-377. The serial numbers of the reference were changed, too.

A list of corrections beyond the comments of the editor:

Line 38: widely distributing across tropical and subtropical regions [1-3]. Due to insufficient genetic

Correction: …distributed…

Line 53: ORG.asm (https://git.metabarcoding.org/org-asm/org-asm), plastome-based evolutionary research have become easier and more cost-effective [22-24].

Correction: …has…

Line 57: results of which were obvious inconsistent with phylogenies based on the nuclear genome.

Correction: …obviously…

Line 60-61: Even so, the chloroplast framework of Brunn-Lund et al. have been replicated by subsequent research generally with an extended dataset [17].

Correction: …has…

Line 73: several controversial taxa in 1980s [1, 34-37] (such as F. dinganensis, F. guizhouensis, and F.

Correction: …in the 1980s…

Line 78: relationship between the complex and these previously described species in 1980s. However, because

Correction: …in the 1980s…

Line 156: samples from the Olmedieae tribe were chosen as the outgroup, according to the previous studies [26, 52].

Correction: …according to previous…

Line 183: sarmentosa var. nipponica and F. sarmentosa var. thunbergii). The CDS regions ranged from 79,149

Correction: Delete one extra space.

Line 234: 60 bp in length was the least, appearing in only once. The maximum length among all the long repeat

Correction: …were…appearing only…

Line255: times. The marjority of the remainding motifs appeared only once to four times, with the exception of

Correction: …majority…remaining…

Line 290-291: exhibited the highest nucleotide diversity: 0.00535, 0.00462, 0.00456, 0.00427 and 0.00377, respectively (Fig 6).

Correction: …0.00427, and 0.00377…

Line 311: signals of positive selection. An infinite Ka/Ks ratios (Ka > 0 and Ks = 0) existed in 27 genes (Fig 7).

Correction: …ratio…

Line364: genomes from the F. sarmentosa complex. Overall, the length of both whole-genome and quadripartite

Correction: …lengths…

Line 404: The plastomic phylogeny of the genus Ficus and the phylogenetic

Done. …plastome…

Line 410: mysterious displacment of certain individuals (F. fulva, F. magnoliifolia, F. albipila, F. albert-smithii, and F. pumila).

Correction: …displacement…

Line 438: rice [29], willow [68], and Orchardgrass [86]. Taken together, the chloroplast genome appears

Correction: …orchardgrass…

---

## [Editor Report · Decision Letter 3]

15 Dec 2022

Comparison of chloroplast genomes and phylogenomics in the Ficus sarmentosa complex (Moraceae)

PONE-D-22-13870R3

Dear Dr. Yao,

We’re pleased to inform you that your manuscript has been judged scientifically suitable for publication and will be formally accepted for publication once it meets all outstanding technical requirements.

Kind regards,

Elvira Hörandl

Academic Editor

PLOS ONE
---

## [Editor Report · Acceptance letter]

22 Dec 2022

PONE-D-22-13870R3 

Comparison of chloroplast genomes and phylogenomics in the *Ficus sarmentosa* complex (Moraceae) 

Dear Dr. Yao:

I'm pleased to inform you that your manuscript has been deemed suitable for publication in PLOS ONE. Congratulations! Your manuscript is now with our production department. 

Kind regards, 

on behalf of

Dr. Elvira Hörandl 

Academic Editor

PLOS ONE